# Sustainability in Rainfed Maize (*Zea mays* L.) Production Using Choice of Corn Variety and Nitrogen Scheduling

**Aakash** [1,2,*], **Narendra Singh Thakur** [1], **Manoj Kumar Singh** [2,*], **Lalita Bhayal** [1], **Kamlesh Meena** [2,3], **Sharad Kumar Choudhary** [1], **Narendra Kumawat** [1], **Ram Kumar Singh** [2], **Udai Pratap Singh** [2], **Shrish Kumar Singh** [4], **Pratik Sanodiya** [2], **Ajay Kumar** [2] and **Anurag Kumar Singh** [2]

[1] Department of Agronomy, Rajmata Vijayaraje Scindia Krishi Vishwa Vidyalaya, College of Agriculture, Indore 452 001, Madhya Pradesh, India; thakur.nst59@gmail.com (N.S.T.); lalitabhayal12567@gmail.com (L.B.); dr.sharad.786@gmail.com (S.K.C.); kumawatandy@gmail.com (N.K.)

[2] Department of Agronomy, Institute of Agricultural Sciences, Banaras Hindu University, Varanasi 221 005, Uttar Pradesh, India; kamalagromony@gmail.com (K.M.); proframkumarsinghciagrbhu@gmail.com (R.K.S.); upsingh61@bhu.ac.in (U.P.S.); prsanodiya10@gmail.com (P.S.); ajaypandey.karwi@gmail.com (A.K.); anuragrau@gmail.com (A.K.S.)

[3] Krishi Vigyan Kendra (Indian Council of Agricultural Research-Indian Institute of Vegetable Research), Deoria 274 506, Uttar Pradesh, India

[4] Department of Agronomy, Tilak Dhari Post Graduate College, Affiliated to Veer Bahadur Singh Purvanchal University, Jaunpur 222 002, Uttar Pradesh, India; tdcagro65@gmail.com

* Correspondence: akash.agro10@bhu.ac.in (A.); mksinghagro@bhu.ac.in (M.K.S.)

**Abstract:** Interestingly more than 50% of the world's area is rainfed and approximately 80% of maize is cultivated under rainfed condition where selection of cultivar and management of nitrogen have major impact on production. The aim of this study was to evaluate the growth, phenology, yield and quality parameters of maize as influenced by variety and nitrogen scheduling under rainfed condition. For this, a field experiment having two factors was laid out in a factorial randomised block design and replicated three times. The first factor was variety, i.e., $V_1$ (JM 216) and $V_2$ (JM 218), and the second was six nitrogen scheduling, i.e., $N_1$ to $N_6$, in which nitrogen splitting was done based on 30-years of average rainfall data. Variety JM 218 and $N_5$ [40 kg N as basal followed by (*fb*) 2 splits of 40 kg N and 38.8 kg N at 30 and 52 days after sowing (DAS) and 1% N foliar spray at 40 DAS] nitrogen scheduling were found promising under rainfed situation because it recorded maximum value of growth parameters, yield attributes, grain yield and quality parameters (protein, mineral and dickson quality index). Thus, it can be inferred that JM 218 and $N_5$ nitrogen schedule would be a better choice than alternative options.

**Keywords:** corn variety; rainfed; N scheduling; N foliar spray; dickson quality index

## 1. Introduction

The "Queen of Cereals", maize (*Zea mays* L.), is the world's third-largest cereal crop. The multiple uses of maize as a food, fodder, feed and more recently fuel has further made it a more demand friendly and a high-value crop. Maize is the basis for food security in some of the world's poverty aligned regions of Africa, Asia and Latin America [1]. Globally, 1148.48 million metric tonnes of maize was harvested in 2019 from 197.20 million hectares of land [2]. About 73 percent of this area is located in the developing world. Maize contributes a significant portion of the food consumed by poor communities in developing countries, yet its production is insufficient to meet the requirement of poor people in these areas.

The demand of maize will be doubled by 2050 in the developing world as per Consultative Group on International Agricultural Research (CGIAR) [3]. According to the FICCI-PwC [4] report, India would require 45 million tonnes of maize output by 2022, with 30 million tonnes necessary for feed and 15 million tonnes required for food, seed, and

industrial applications. This means that output must rise at a compound annual growth rate (CAGR) of 15% to fulfill its demand.

The term "rainfed agriculture" is used to describe the farming practice that relies on rainfall for water [5]. Rainfed agriculture is the most important sector for providing food security [6]. Despite the fact that rainfed crop production is becoming more unpredictable, 90% of maize farmers in rural agricultural areas rely on rainfed maize production as a source of income [7]. Globally, more than 50% of total cultivated area of maize as a share of their total cultivated land is under rainfed conditions [8] except in Oceania, thus, effective utilization of these areas for enhancing maize production are the necessity for future existence. The irregular or uneven distribution patterns of monsoon rains in South East Asia is the prime feature of rainfed which can produce intermittent moisture stress at various crop growth stages, therefore moisture availability is rarely enough for rainfed maize. All the nutrients are absorbed by plant root after its ionic form dissolve in soil solution. Thus, nutrients and water shortage are the major limiting factors for the low productivity of rainfed maize, often locally known as *Kharif* maize [9]. Furthermore, because of the uncertain return, farmers are hesitant to invest in better seed/variety and other inputs, resulting in poor yields. In many areas, increasing population has placed substantial pressure on rainfed cropland. The challenges of low and depleting essential nutrients and organic matter in soils are prevalent on rainfed croplands [10]. Moreover, due to the fact of expanded climate variability, climate change is anticipated to make rainfed farmers more vulnerable to local weather change [11].

Selection of location specific variety is one of the most essential agronomic activities [12]. Variety change has played a key role in improving maize productivity, according to Chen et al. [13], with the contribution of variety to yield rising from 21.0% to 44.3% during the last 50 years. Currently, many varieties of maize have been evolved and each needs specific management practices and climatic requirements on which it reaches its full genetic potential. Therefore, a comparison of varieties for growth and yield characteristics under various nutrient management regimes is necessary [14]. Foremost important among them is careful management of nitrogen because on the one hand, it is the most imperative element for proper growth and development of plants and so the other is the global challenge of feeding world's ever increasing population that would be impossible without nitrogen fertilizer since it increases the production and profitability of every individual farmer. However, it's over use in many cases causes pollution of rivers, lakes and coastal water around the world, and contributes to the emission of greenhouse gases and eutrophication. So excessive application is wast of money and needlessly worsening environmental problems [15]. Molden [16] said that rapid improvements in rainfed yields in some places in recent years can be attributed to better fertiliser management and selection of appropriate variety. According to Bindhani et al. [17] applying nitrogen in three equal portions at the time of sowing, knee high stage, and pre-tasseling stage improved plant height, dry matter accumulation, leaf area and yield of maize. Afifi et al. [18] pointed out that 100% soil application nitrogen rate combined with 0.7% foliar application of nitrogen; produced maximum maize grain yield, while according to Khan et al. [19] 2% N foliar spray along with soil application is a useful strategy to get improved yield of maize.

Therefore, it is important to assess the magnitude of genotype response, to nitrogen scheduling especially under rainfed condition, since the major reason for low productivity of maize is mainly abiotic stress, as approximately 80% area of maize is grown under rainfed conditions [20] where climatic fluctuations especially rainfall abnormalities and temperature variations cause great loss of nitrogen via leaching, denitrification, volatilization, surface runoff. In addition to these losses, lack of rainfall affects the nutrient uptake from the soil which finally declines; nitrogen use efficiency (NUE) and performance of maize varieties; thus, ultimately reducing productive potential of crop. In order to explain the hypothesis of present experiment more clearly Figure 1. has been inserted conceptualizing how the 30-years long-term rainfall data and choice of location specific variety improved nitrogen nutrition.

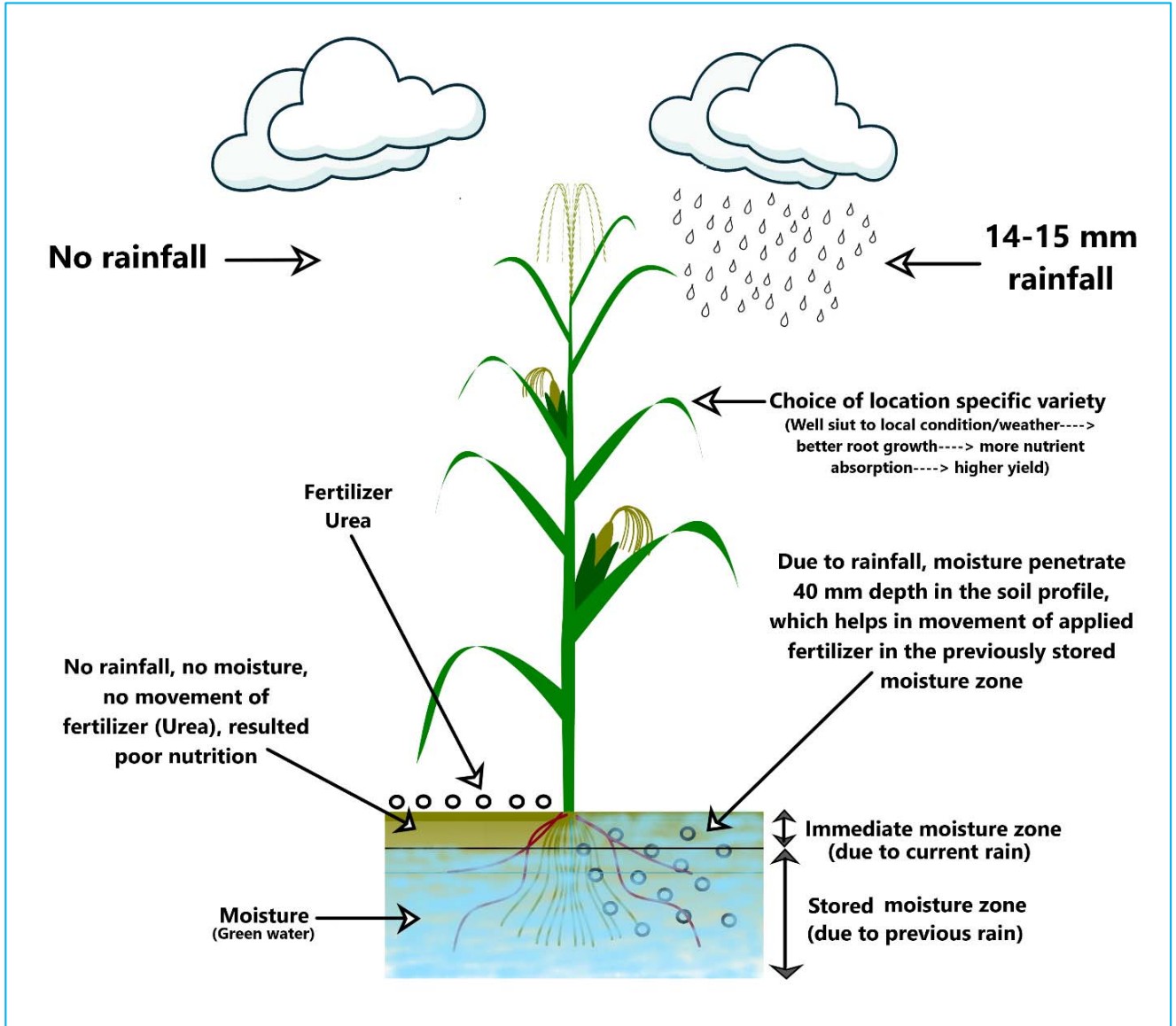

**Figure 1.** Hypothesis of experiment is conceived with 30-years long-term meteorological data available for the site and the crop was raised under rainfed condition. Moisture from top 35–40 mm of soil depth evaporates quickly and soil becomes dry when dry spell is prolongs, however, soil below the 35–40 mm of depth had variable soil profile moisture (depicted in Figure 1 and termed as 'stored moisture zone'). Fertilizer (Urea) in the absence of moisture in the topsoil, does not go inside the soil and losses through volatilization (depicted in Left side portion of Figure 1). In the present study nitrogen is scheduled based on 30-years average rainfall data when 14–15 mm rainfall is expected to be received which moistened top 40 mm of soil profile of the experimental soil i.e., clayey soil (in figure termed as 'top moisture zone'). Therefore, applied fertilizer moves all round in the top moisture zone as well as meets with 'stored moisture zone' of soil and becomes available to plant roots, (depicted in the Right side portion of Figure 1) leading to better nutrition and yield.

By considering these challenges in rainfed conditions, an experiment was planned through evaluation of varieties with nitrogen scheduling based on 30-years average rainfall data. The main objectives of study were to find out the suitable nitrogen schedule and variety for improving growth, yield and quality parameters of maize under rainfed condition.

## 2. Materials and Methods

### 2.1. Experimental Site, Climatic Condition and Soil Property

The experiment was conducted at Research Farm of Rajmata Vijayaraje Scindia Krishi Vishwa Vidyalaya, College of Agriculture, Indore, Madhya Pradesh, India (Figure 2) during *Kharif* season (2018–19). The topography of field was uniform with gentle slope. Indore is situated at an altitude of 555.5 m above mean sea level (MSL). It is located at latitude 22.43° N and longitude of 75.66° E. This region enjoys sub-tropical, semi-arid type climate. The mean minimum and maximum temperature ranges from 7 °C to 23 °C and 23 °C to 43 °C, in winter and summer season, respectively. The monsoon activities during the experimental year had commenced in the 22nd Standard Meteorological Week (SMW) and continued till the 38th SMW and during crop growth period (27th SMW to 43rd SMW) 685 mm rain was received in 34 rainy days (Figure 3). The soil of experimental site was predominantly clayey in texture, slightly alkaline in reaction (pH 7.60) and low in organic carbon (0.40%) and available nitrogen (188 kg·ha$^{-1}$), medium in available phosphorus (15.8 kg·ha$^{-1}$) and high in available potash (526 kg·ha$^{-1}$) (Table 1).

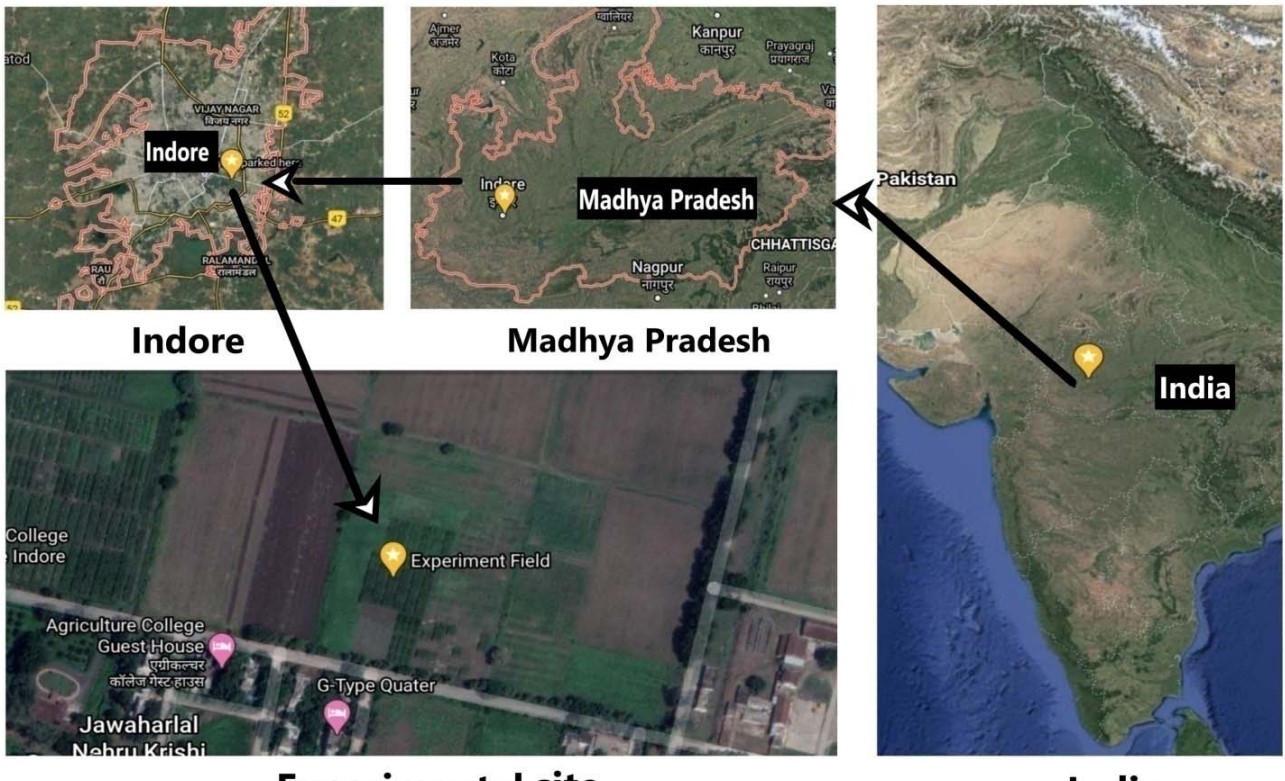

**Figure 2.** Location of the experimental site.

**Table 1.** Physico-chemical properties of the soil of the experimental field.

| Mechanical Analysis Values of Soil | | | |
|---|---|---|---|
| S. No. | Soil Particle | Quantity (%) | Method of Analysis |
| 1 | Sand | 12.43% | |
| 2 | Silt | 38.03% | Hydrometer method [21] |
| 3 | Clay | 49.54% | |
| 4 | Textural class | Clay soil | - |

**Table 1.** *Cont.*

| | Chemical Analysis Values of Soil | | |
|---|---|---|---|
| **S. No.** | **Analysis** | **Values** | **Method Adopted** |
| 1 | Soil pH | 7.60 | Glass electrode method [22] |
| 2 | Electrical conductivity (dS·m$^{-1}$) | 0.26 | [22] |
| 3 | Organic carbon (%) | 0.40 | [23] |
| 4 | Available nitrogen (kg·ha$^{-1}$) | 188 | Alkaline permanganate method [24] (estimates ammonium nitrogen) |
| 5 | Available phosphorus (kg·ha$^{-1}$) | 15.8 | Olsen's method [25] |
| 6 | Available potassium (kg·ha$^{-1}$) | 526 | Flame photometer [26] |

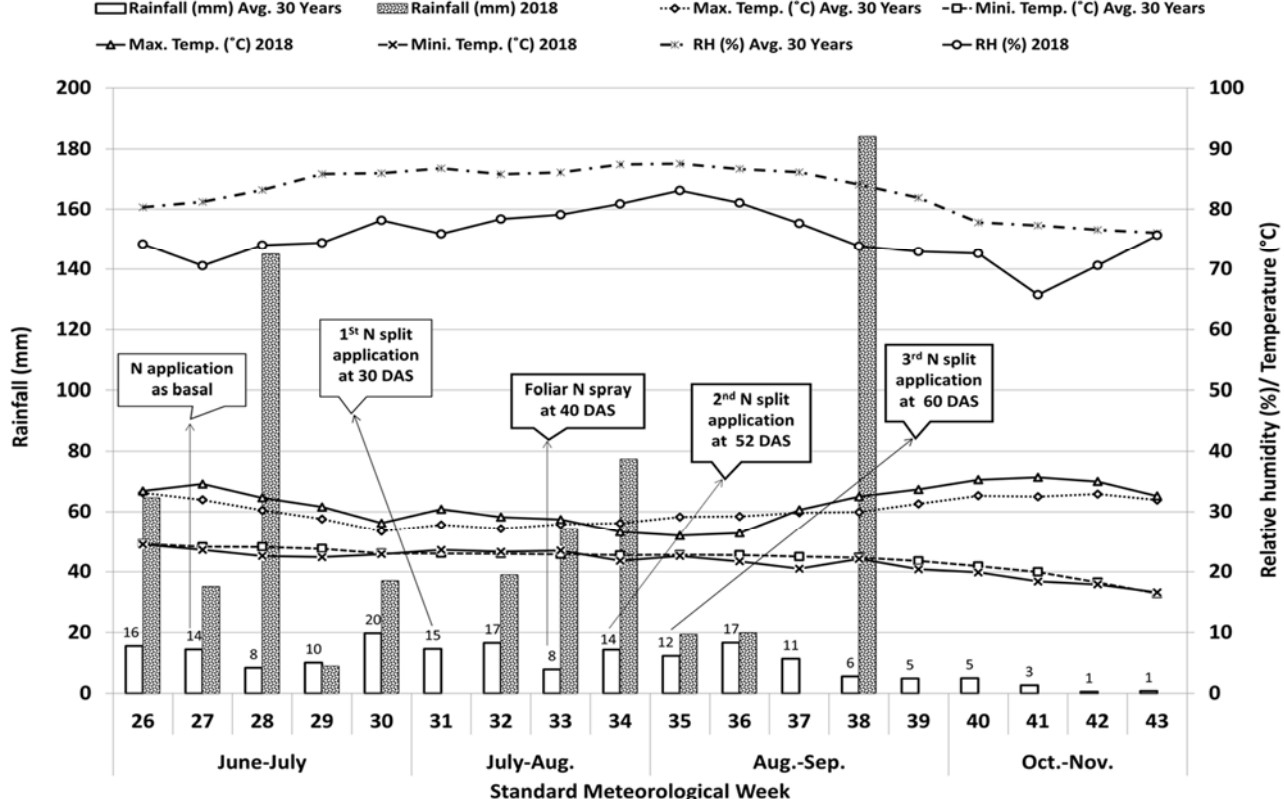

**Figure 3.** Rainfall and other weather parameters during crop growing period of experimental crop season and 30-years average.

### 2.2. Experimental Set Up and Treatment Details

The experiment consisted of two factors which made twelve treatment combinations laid out in factorial randomized block design and replicated thrice. The first factor was variety which consisted two levels, i.e., $V_1$ (JM 216) and $V_2$ (JM 218) and the second factor was nitrogen scheduling based on 30-years average rainfall data which consisted of six levels presented in Table 2. Nitrogen splitting was scheduled on the basis of 30-years average rainfall data in the Standard Meteorological Week (SMW) receiving rainfall in the range of 14–15 mm (Figure 3). This amount of rainfall was found to recharge soil profile moisture sufficient to make nitrogenous fertilizer available to maize crop. In this regards after the basal application of nitrogen (N), 1st split of N was applied at 30 DAS, 2nd split of N applied at 52 DAS and 3rd split of N applied at 60 DAS. The image of the experiment is shown in Figure 4, which shows the views of crop at initial and 35 days after sowing (DAS) and impact of $N_5$ treatment on JM 218.

**Table 2.** Treatment details.

| Varity (V) | |
|---|---|
| **Varity (V)** | |
| V$_1$ | JM 216 |
| V$_2$ | JM 218 |
| **Nitrogen Scheduling Based on 30-Years Average Rainfall Data (Ns30RF)** | |
| N$_1$ | 40 kg N as basal *fb* 2 splits of 40 kg N and 30 kg N at 30 and 52 DAS |
| N$_2$ | 60 kg N as basal *fb* 2 splits of 30 kg N at 30 and 52 DAS |
| N$_3$ | 30 kg N as basal *fb* 2 splits of 60 kg N and 30 kg N at 30 and 52 DAS |
| N$_4$ | 30 kg N as basal *fb* 3 splits of 30 kg N at 30, 52 and 60 DAS |
| N$_5$ | 40 kg N as basal *fb* 2 splits of 40 kg N and 38.8 kg N at 30 and 52 DAS and 1% N foliar spray at 40 DAS |
| N$_6$ | 30 kg N as basal *fb* 3 splits of 30 kg N, 30 kg N and 28.8 kg N at 30, 52 and 60 DAS and 1% N foliar spray at 40 DAS |

Abbreviations: N = nitrogen; *fb* = followed by; DAS = days after sowing; All doses were applied per hectare basis.

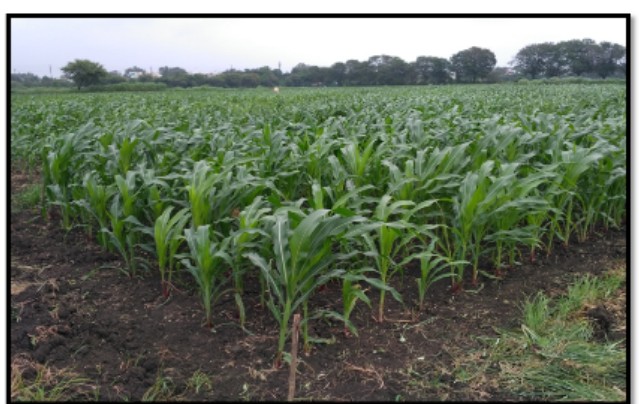

**Crop at initial stage**

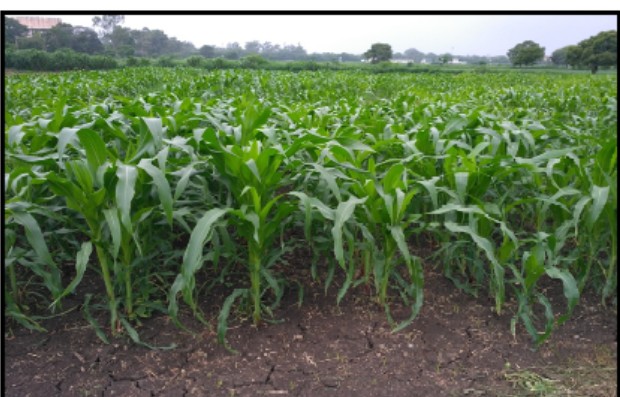

**Crop at 35 DAS**

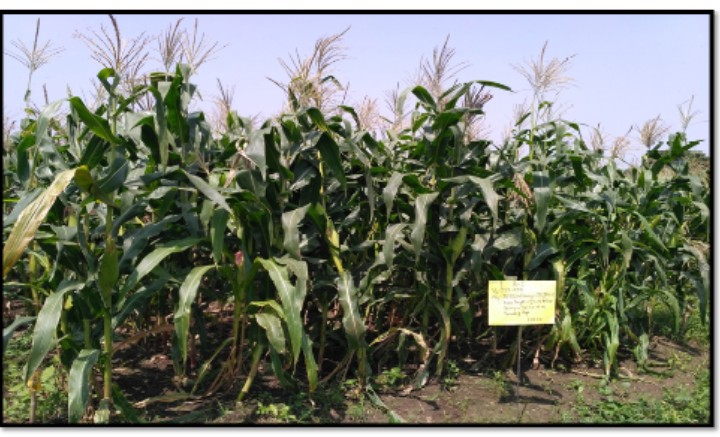

**View of JM 218 with N$_5$**

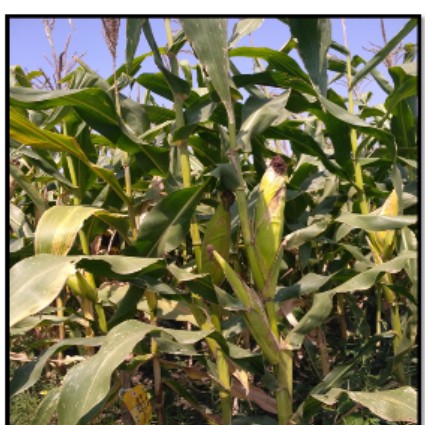

**Cob of JM 218**

**Figure 4.** Views of experiment.

*2.3. Crop Management*

Ploughing was done for primary tillage and then a suitable flat seedbed was prepared by giving harrowing with tractor drawn harrow followed by planking. Thereafter, the experiment was laid out as per plan and design. To ensure the good germination, healthy and good quality seeds were used. The seeds were treated by Carbendazim at the rate of 3 g per kg of seed. Sowing was done by using seeds at the rate of 12 kg·ha$^{-1}$ through dibbling method with 60 cm row to row and 22 cm plant to plant spacing to obtain 75,757 plants·ha$^{-1}$. Nutrients were applied at the rate of 120–50–30; N-P$_2$O$_5$-K$_2$O kg·ha$^{-1}$.

Entire dose of $P_2O_5$ and $K_2O$ were applied as basal using Diammonium Phosphate (DAP) and Muriate of Potash (MOP), respectively and the total nitrogen amount (120 kg·ha$^{-1}$) was split into various amounts as per treatment which was applied as soil (DAP + Urea), broadcasted in standing crop and foliar application (urea only) as per last 30-years average rainfall data (Table 3). Hand weeding was done once at 20 days after sowing (DAS) and spray of herbicide Topramezone @30 g active ingredient·ha$^{-1}$ in later stage of crop in order to control late emerging weeds. For the control of stem borer, a pinch of Carbofuran 3G granules at 35 DAS was placed in leaf whorls since first instar caterpillars of stem borer congregate in leaf whorls. The crop of net plot area (5.0 m × 3.6 m) was harvested along with cob when plant turned golden yellow with the help of sickle. The crop was harvested close to ground and plant were tied in bundles and kept for sun drying on threshing floor for seven days. The threshing was done using maize thresher.

**Table 3.** Summary of application of nitrogen doses.

| Treatment Symbols | Nitrogen Quantity (kg N ha$^{-1}$) | | | | |
|---|---|---|---|---|---|
| | As Basal (At 0 DAS) | At 30 DAS | At 40 DAS * | At 52 DAS | At 60 DAS |
| N$_1$ | 40.0 | 40.0 | - | 40.0 | - |
| N$_2$ | 60.0 | 30.0 | - | 30.0 | - |
| N$_3$ | 30.0 | 60.0 | - | 30.0 | - |
| N$_4$ | 30.0 | 30.0 | - | 30.0 | 30.0 |
| N$_5$ | 40.0 | 40.0 | 1.2 | 38.8 | - |
| N$_6$ | 30.0 | 30.0 | 1.2 | 30.0 | 28.8 |

* Applied as foliar spray via low volume sprayer (1.2 kg N dissolved in 120 lit of water).

### 2.4. Sampling and Observations

Investigations on plant growth characteristics i.e., plant height, leaf area, leaf area index, dry matter and crop growth rate were carried out. For these, five plants from each net plot area (5.0 m × 3.6 m) were selected randomly and tagged for observations. The mean of various parameters of these five tagged plants were computed and used for statistical analysis.

### 2.4.1. Dry Matter (g·plant$^{-1}$)

The dry matter accumulation by crop was recorded at harvest by destructive method. First the samples were sun-dried and then dried in the oven at 68 °C ± 2 °C till constant weight was obtained and then dried samples were weighed separately using by electronic balance.

### 2.4.2. Leaf Area (cm$^2$·plant$^{-1}$)

Length (L) of all the fully opened leaf lamina per plant was measured from the base to the tip of the leaf. Breadth (B) was taken at three portions i.e., below the middle, middle and above the middle portion of the leaf lamina in order to get average breadth. The products of leaf length and breadth were multiplied by the factor 0.75 [27].

$$\text{Leaf area} = L \cdot B \cdot 0.75$$

### 2.4.3. Leaf Area Index (LAI)

The leaf area of five randomly selected plants was recorded and thereafter, it was divided by land area to obtain leaf area index (LAI). It was determined by using following formula suggested by Watson [28]:

$$\text{LAI} = \text{Total leaf area (m}^2\text{)}/\text{Land area (m}^2\text{)}$$

### 2.4.4. Crop Growth Rate (g·m$^{-2}$·day$^{-1}$)

The crop growth rate was calculated as an increase in dry matter production per unit ground area per unit time. In this investigation the crop growth rate was worked out using following formula [28].

$$\text{CGR} = \{ \, (W_2 - W_1)/(t_2 - t_1) \cdot P \, \}$$

where, P is ground area and $W_1$ and $W_2$ are the total dry weight at time $t_1$ and $t_2$, respectively.

### 2.4.5. Crop Phenology

Daily observations of emergence were recorded from each net plot area, starting from two days after sowing. Five plants per net plot area were randomly tagged and observations were made on daily basis in order to determine six and fifteenth leaf stage. The number of days to tasseling and silking stage were recorded by counting from sowing to date when 50% tassels and silks in each net plot area emerged. Days to physiological maturity was also counted from sowing to till 20–30% of plant and husk turn to golden yellow and physical enlargement of cob was stopped.

### 2.4.6. Dickson Quality Index (DQI)

It is a quality parameter related to morphology of plants, which determines survival of plant during its initial stage of life cycle. DQI is calculated from below equation suggested by Dickson et al. [29].

$$\text{DQI} = \text{Total dry matter (g)}/\{ \, [\text{PH/SD}] + [\text{AGDM/RDM (g)}] \}$$

where, PH is plant height (cm), SD is stem diameter (mm), AGDM is above ground dry matter mass (g) and RDM is root dry matter mass (g).

### 2.4.7. Yield Attributes

Five cobs were randomly selected from each net plot area to evaluate cob length, number of grain rows·cob$^{-1}$, and number of grains·cob$^{-1}$. The cob length was measured with the help of cm scale, while other parameters were recorded by counting and the treatment mean values were subjected to statistical analysis. For 1000 grain weight, all the cobs from each net plot were thrashed and one thousand grains were counted from the yield of each net plot and then weighed.

### 2.4.8. Yield (kg·ha$^{-1}$)

Cobs harvested from each net plot were sun-dried for seven days before being threshed with a maize thresher and then grain output was recorded. The grain yield was adjusted to a moisture content of 12%. After removing the cobs, leftover plant material, including the husk, was sun dried and weighed for stover yield.

### 2.4.9. Protein Content

Firstly, total nitrogen concentration (%) in the grain was determined using by Kjeldahl method and then value of protein concentration was obtained by multiplying the nitrogen concentration with a factor 6.25 [30].

$$\text{Protein content (\%)} = \text{N percent} \cdot 6.25$$

### 2.4.10. Total Mineral

A grain sample of 5 g was grounded and taken in pre-weighed crucibles and was burnt until charred. The crucibles with charred material were placed in a muffle furnace at

about 550 °C until greyish—white residue was obtained. From the weight of the residue, the percentage of ash was calculated as following:

$$\text{Total minerals }(\%) = \{\text{Weight after ignition (g)/Weight of sample (g)}\} \cdot 100$$

### 2.5. Statistical Analysis

The data obtained from growth, yield attribute, yield and quality parameters were subjected to analysis of variance (ANOVA) using Statistical Tool for Agricultural Research (STAR) software (version STAR 2.0.1, IRRI, Los Baños, Philippines), while the significance of differences between treatment mean values was determined using the Tukey's HSD (honestly significant difference) test at 5% and 1% levels.

## 3. Results and Discussion

As shown in results maize growth, yield components, yield and quality parameters were significant ($p \leq 0.05$) or highly significant ($p \leq 0.01$) affected by variety (V) and nitrogen scheduling based on 30-years average rainfall data (Ns30RF) whereas their interactions (V × Ns30RF) were found to be statistically similar.

### 3.1. Growth Parameters

The results presented in the Table 4 on the growth parameters elucidated that the maize variety JM 218 was found to be significantly superior ($p \leq 0.05$) to JM 216 in terms of plant height, dry matter (DM), crop growth rate (CGR), leaf area and leaf area index (LAI). When compared to JM 216; plant height, DM, CGR, leaf area and LAI values of JM 218 increased by 8.61%, 15.65%, 13.73%, 10.58% and 9.13%, respectively. The differences in plant growth attributes between two varieties might be due to the genetical modified and/or by environmental factors resulting in morphological variations and varied nitrogen required for the synthesis of different chemical constituents at different plant organ levels. Nabila et al. [31] found that maize cultivar National 6 reported maximum plant height, DM, leaf area, LAI and LAR at 70 DAS as compared to cultivar T.W. 329, due to morphological differences in variety. It was also observed that JM 218 variety performed better than JM 216 since it had more leaf breadth and length which helped in capturing more solar radiation leading to more crop growth rate and dry matter accumulation. Abera et al. [32] reported that variations in leaf size of different varieties of maize produced significant differences in leaf area, LAI and CGR. Amin et al. [33] and Hassanein et al. [34] also found similar findings.

**Table 4.** Effect of variety and nitrogen scheduling based on 30-years average rainfall data on growth attributes.

| Treatment | Growth Attributes | | | |
|---|---|---|---|---|
| | **Plant Height (cm)** | **CGR ($g \cdot d^{-1} \cdot m^{-2}$)** | **Dry Matter ($g \cdot plant^{-1}$)** | **Leaf Area ($cm^2 \cdot plant^{-1}$)** |
| | **At Harvest** | **50–75 DAS** | **At Harvest** | **75 DAS** |
| **Variety (V)** | | | | |
| V$_1$-JM 216 | 184.5 [b] | 33.5 [b] | 290.6 [b] | 5929.8 [b] |
| V$_2$-JM 218 | 200.4 [a] | 38.1 [a] | 336.1 [a] | 6557.6 [a] |
| *HSD* ($p \leq 0.05$) | 13.4 | 2.4 | 18.1 | 373.6 |
| **Nitrogen Scheduling Based on 30-Years Average Rainfall Data (Ns30RF)** | | | | |
| N$_1$-40 kg N as basal *fb* 2 splits of 40 kg N/ha and 30 kg N/ha at 30 and 52 DAS | 188.4 [abc] | 34.9 [bc] | 306.8 [ab] | 6128.5 [abc] |
| N$_2$-60 kg N//ha as basal *fb* 2 splits of 30 kg N/ha at 30 and 52 DAS | 180.5 [bc] | 29.7 [c] | 297.6 [ab] | 5764.0 [bc] |

**Table 4.** *Cont.*

| Treatment | Growth Attributes | | | |
|---|---|---|---|---|
| | Plant Height (cm) | CGR (g·d$^{-1}$·m$^{-2}$) | Dry Matter (g·plant$^{-1}$) | Leaf Area (cm$^2$·plant$^{-1}$) |
| | At Harvest | 50–75 DAS | At Harvest | 75 DAS |
| **Nitrogen Scheduling Based on 30-Years Average Rainfall Data (Ns30RF)** | | | | |
| N$_3$-30 kg N/ha as basal *fb* 2 splits of 60 kg N/ha and 30 kg N/ha at 30 and 52 DAS | 177.2 [c] | 29.3 [c] | 290.2 [b] | 5598.1 [c] |
| N$_4$-30 kg N/ha as basal *fb* 3 splits of 30 kg N/ha at 30, 52 and 60 DAS | 195.8 [abc] | 37.3 [ab] | 316.7 [ab] | 6411.2 [abc] |
| N$_5$-40 kg N/ha as basal *fb* 2 splits of 40 kg N/ha and 38.8 kg N/ha at 30 and 52 DAS and 1% N foliar spray at 40 DAS | 209.8 [a] | 42.2 [a] | 340.3 [a] | 6933.0 [a] |
| N$_6$-30 kg N//ha as basal *fb* 3 splits of 30 kg N/ha, 30 kg N/ha and 28.8 kg N/ha at 30, 52 and 60 DAS and 1% N foliar spray at 40 DAS | 203.0 [ab] | 41.5 [a] | 328.7 [ab] | 6627.4 [ab] |
| *HSD* ($p \leq 0.05$) | 23.2 | 6.24 | 46.9 | 972.03 |
| Source of variation | | | | |
| V | * | * | * | * |
| Ns30RF | * | ** | * | ** |
| V × Ns30RF | ns | ns | ns | ns |

Different letters, i.e., [a, b, c] within a column indicate significant differences according to honestly significant difference (HSD) test ($\alpha$ = 0.05). Significance levels: * $p \leq 0.05$; ** $p \leq 0.01$; ns, not significant ($p > 0.05$).

Under the effect of Ns30RF, N$_5$ (40 kg N as basal *fb* 2 splits of 40 kg N and 38.8 kg N at 30 and 52 DAS and 1% N foliar spray at 40 DAS) responded better in terms of plant height (209.8 cm), accumulation of maximum DM (340.3 g·plant$^{-1}$), maximum leaf area production (6933.0 cm$^2$·plant$^{-1}$), higher LAI (5.25) (Figure 5) and achieved fastest CGR (42.2 g·d$^{-1}$·m$^{-2}$) which was found to be statistically at par with N$_6$. This might be due to split applications of nitrogen at 30 and 52 DAS (31 and 34 Standard Meteorological Week) and as per the 30-years long term average data 14–15 mm rainfall was expected to be received during these standard weeks which were enough to recharge 40 mm of soil profile moisture. However, in present experiment 37 mm and 77 mm rainfall was received in 3 days and 4 days before 1st and 2nd split application, respectively and at that time 70 mm and 122 mm soil moisture, respectively was available in soil profile which resulted in optimum availability of nitrogen to crop plants at their actively nitrogen consuming phase which might have brought more cell division and expansion, chlorophyll formation (increases photosynthetic rate) and vegetative growth; resulting in increase of plant height, DM, leaf area, LAI and CGR. These findings are almost in line with Afifi et al. [18] who reported that 100% RDN applied in soil under optimal soil profile moisture and supplemented with urea foliar application significantly increased growth parameters of maize. Bindhani et al. [17] also reported that the application of nitrogen in three equal splits i.e., 1/3rd at the time of sowing, 1/3rd at knee high and 1/3rd at pre tasseling stage, significantly increased plant height, dry matter accumulation and leaf area. Similar results were also reported by Hammad et al. [35], Wasaya et al. [36], Sharifi and Namvar [37] and Verma et al. [38].

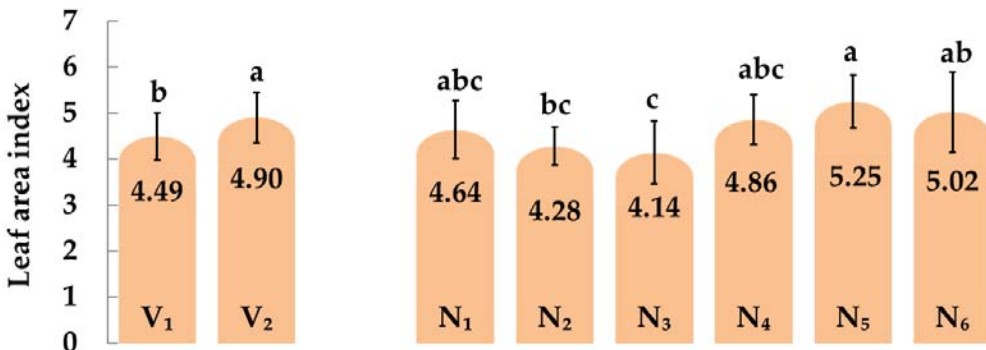

**Figure 5.** Effect of variety and nitrogen scheduling based on 30-years average rainfall data on leaf area index. Different letters, i.e., [a, b, c] indicate significant differences according to honestly significant difference (HSD) test ($\alpha$ = 0.05). Significance levels: * $p \leq 0.05$.

*3.2. Crop Phenology*

In this study, the apparent significant variations in the crop development stages were observed except days to emergence (Figure 6). From sowing (S) to emergence (VE) Ns30RF did not imposed better effect, while the variations in S to VE for both varieties were between 1 and 2 days. JM 218 took fewer days (24 days) than JM 216 (28 days) to go from VE to the six leaf stage (V6). JM 216 reached the fifteen leaf stage (V15) (48 days) and silking (R1) stage (61 days) later than JM 218 (46 and 58 days, respectively). Despite this, JM 216 needed fewer days to reach physiological maturity (PM) (103 days) than JM 218 (106 days). With respect to impact of Ns30RF on phenology, $N_5$ (40 kg N as basal *fb* 2 splits of 40 kg N and 38.8 kg N at 30 and 52 DAS, respectively and 1% N foliar spray at 40 DAS) application had taken maximum days to reach at V6 stage. From V6 to V15, $N_1$ and $N_2$ took maximum duration (18 and 20 days) for JM 216 and JM 218, respectively whereas the minimum days were required under $N_5$. When comparing days to tasseling (VT) (i.e., S to VT) and days to silking (R1) (i.e., S to R1), $N_5$ recorded earlier appearance of tassel (52 and 50) and silk (57 and 54) for JM 216 and JM 218, respectively. More days were taken by $N_5$ between R1 and PM as compared to leftover Ns30RF. Similarly, under $N_5$, the entire crop duration took the more days. Hammad [39] noted that split nitrogen application and varieties showed identical effect for days to emergence. In the present study, V6, V15, VT and R1 stages appeared first in $N_5$ treatment, which indicates that when nitrogen was applied by split doses followed by foliar application then available soil nitrogen matched with crop nitrogen needs, leading to faster crop growth. Adhikari et al. [40] also supported these findings. Anjum et al. [41] reported more days to physiological maturity when nitrogen applied in three equal splits. An increase in split number of N application could maximize photosynthesis rate in plants that might have resulted in increased durability of leaf and it delayed the phenological characteristics in the maize crop [42]. As a general rule a crop with longer duration has more time for photosynthesis and translocation of photosynthates from source (leaf) to sink (grain/economic product) resulting in higher grain yield. Dolan et al. [43] observed higher nutrient availability due to appropriate scheduling of nitrogen fertilizer that could be a possible reason for delayed phenological events. Foliar spray of nitrogen accelerated and prolongs vegetative development which might be the cause of delayed maturity. These findings are also backed up by Khan et al. [19] who found that foliar N spraying at 45 days after emergence delayed maize maturity.

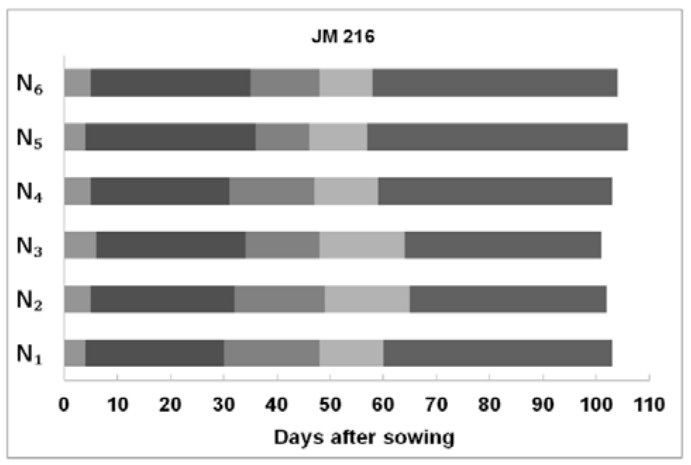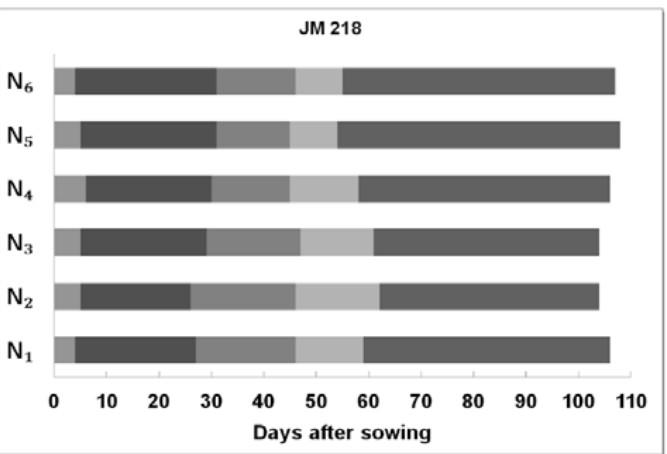

■ S-VE ■ VE-V6 ■ V6-V15 ■ VT-R1 ■ R1-PM

**Figure 6.** Durations of maize varieties as influenced by nitrogen scheduling based on 30-years average rainfall data. S: sowing; VE: emergency stage; V6: sixth leaf stage; V15: fifteenth leaf stage; VT: tasseling stage; R1: silking stage and PM: physiological maturity stages.

### 3.3. Yield Attributes

The outcome of statistical analysis (Table 5) corresponding to yield attributes revealed that under the factor variety; JM 218 registered higher length of cob (20.0 cm), number of grain rows$\cdot$cob$^{-1}$ (14.0), number of grains$\cdot$row$^{-1}$ (39.7) and 1000-grain weight (309.6 g), and was found to be statistically significant over JM 216. The increment in cob length, number of grain rows$\cdot$cob$^{-1}$, number of grains$\cdot$row$^{-1}$ and 1000-grain weight of JM 218 over JM 216 in terms of percentage were 8.10, 12.90, 5.02 and 8.40, respectively. JM 218 developed canopy faster, accumulated more dry matter and had slower leaf senescence than JM 216. These attributes helped in achieving higher cob length, grain rows$\cdot$cob$^{-1}$, grains$\cdot$row$^{-1}$ and 1000-grain weight. Meng et al. [44] and Asaduzzaman et al. [45] research findings are in favor of these findings, as they reported that maintaining a green canopy and delaying leaf senescence by genotype prolongs photosynthesis which in turn contributes to yield attributes. The differences in genotypic yield attributes due to higher leaf area were also observed by Kasikranan et al. [46]. Castro et al. [47] also reported that yield attributes may vary depending on the genotype special ability for some traits.

**Table 5.** Effect of variety and nitrogen scheduling based on 30-years average rainfall data on yield attributes and yields.

| Treatment | Yield Attributes | | | Yields | |
|---|---|---|---|---|---|
| | Length of Cob (cm) | Number of Grain Rows$\cdot$Cob$^{-1}$ | 1000-Grain Weight (g) | Grain Yield (kg$\cdot$ha$^{-1}$) | Stover Yield (kg$\cdot$ha$^{-1}$) |
| **Variety (V)** | | | | | |
| V$_1$-JM 216 | 18.5 [b] | 12.4 [b] | 285.4 [b] | 5685 [b] | 10,338 [b] |
| V$_2$-JM 218 | 20.0 [a] | 14.0 [a] | 309.6 [a] | 6139 [a] | 11,107 [a] |
| *HSD* ($p \leq 0.05$) | 0.8 | 0.8 | 13.5 | 215 | 285 |
| **Nitrogen Scheduling Based on 30-Years Average Rainfall Data (Ns30 RF)** | | | | | |
| N$_1$-40 kg N/ha as basal *fb* 2 splits of 40 kg N/ha and 30 kg N/ha at 30 and 52 DAS | 19.0 [abc] | 12.8 [abc] | 295.1 [abc] | 5883 [ab] | 10,691 [ab] |
| N$_2$-60 kg N//ha as basal *fb* 2 splits of 30 kg N/ha at 30 and 52 DAS | 18.5 [bc] | 12.2 [bc] | 279.2 [bc] | 5754 [ab] | 10,459 [b] |

**Table 5.** *Cont.*

| Treatment | Yield Attributes | | | Yields | |
|---|---|---|---|---|---|
| | Length of Cob (cm) | Number of Grain Rows·Cob$^{-1}$ | 1000-Grain Weight (g) | Grain Yield (kg·ha$^{-1}$) | Stover Yield (kg·ha$^{-1}$) |
| N$_3$-30 kg N/ha as basal *fb* 2 splits of 60 kg N/ha and 30 kg N/ha at 30 and 52 DAS | 17.9 [c] | 11.9 [c] | 269.1 [c] | 5607 [b] | 10,269 [b] |
| **Nitrogen Scheduling Based on 30-Years Average Rainfall Data (Ns30 RF)** | | | | | |
| N$_4$-30 kg N/ha as basal *fb* 3 splits of 30 kg N/ha at 30, 52 and 60 DAS | 19.2 [abc] | 13.5 [abc] | 305.3 [ab] | 5975 [ab] | 10,801 [ab] |
| N$_5$-40 kg N/ha as basal *fb* 2 splits of 40 kg N/ha and 38.8 kg N/ha at 30 and 52 DAS and 1% N foliar spray at 40 DAS | 20.7 [a] | 14.7 [a] | 322.6 [a] | 6197 [a] | 11,207 [a] |
| N$_6$-30 kg N//ha as basal *fb* 3 splits of 30 kg N/ha, 30 kg N/ha and 28.8 kg N/ha at 30, 52 and 60 DAS and 1% N foliar spray at 40 DAS | 20.1 [ab] | 14.2 [ab] | 313.8 [ab] | 6057 [ab] | 10,911 [ab] |
| *HSD* ($p \leq 0.05$) | 2.03 | 2.09 | 35.08 | 558.8 | 742.32 |
| **Source of Variation** | | | | | |
| V | * | * | * | * | * |
| Ns30RF | ** | ** | ** | * | * |
| V × Ns30RF | ns | ns | ns | ns | ns |

Different letters, i.e., [a, b, c] within a column indicate significant differences according to honestly significant difference (HSD) test ($\alpha = 0.05$). Significance levels: * $p \leq 0.05$; ** $p \leq 0.01$; ns, not significant ($p > 0.05$).

In case of Ns30RF, N$_5$ recorded significant variations in terms of maximum cob length (20.7 cm), more number of grain rows (14.7·cob$^{-1}$), more number of grains (41.4·row$^{-1}$) (Figure 7) and higher 1000-grain weight (322.6 g), and was comparable to N$_6$. This specifies that split application of nitrogen supplemented by foliar application of nitrogen improve nitrogen nutrition resulting in realization of better yield attributes. These results corroborated the findings of Sharifi and Namvar [37], Nabila et al. [31] and Afifi et al. [18] who found that the application of N in three equal splits produced higher cob length, number of grain rows·cob$^{-1}$, number of grains·row$^{-1}$, number of grains·cob$^{-1}$ and 1000-grain weight. Similar results were also reported by Nemati and Sharifi [48].

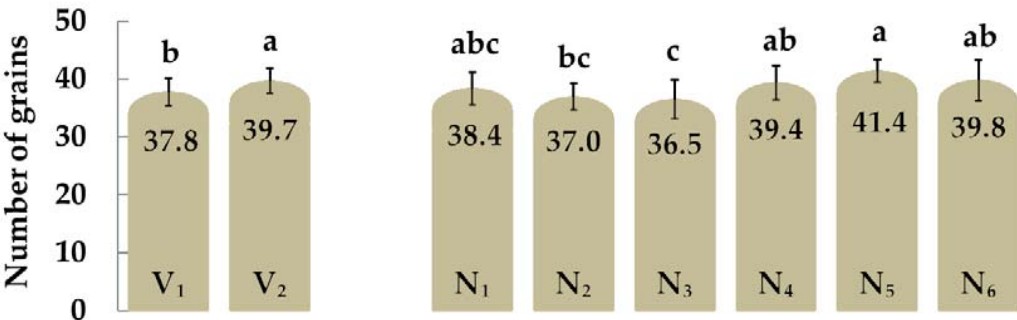

**Figure 7.** Effect of variety and nitrogen scheduling based on 30-years average rainfall data on number of grains·row$^{-1}$. Different letters, i.e., [a, b, c] indicate significant differences according to honestly significant difference (HSD) test ($\alpha = 0.05$). Significance levels: * $p \leq 0.05$.

*3.4. Yield of Maize*

JM 218 recorded 7.99% more grain yield and 7.44% more stover yield compared to JM 216 and proved significantly superior to JM 216. These variations in yield might be probably due to the higher leaf area production by JM 218 which helps in more solar radia-

tion capture resulting in more photosynthetic activity and dry matter accumulation, and longer maturity which also enhance grain filling period as well as helps in photosynthates translocation into the grains, as compared to JM 216. Similar trend with varieties was observed by Nwogboduhu [49] who found that the differences in yield production ability of cultivars i.e., Sammaz 17 grain yield (4.35 t·ha$^{-1}$) as compared to Sammaz 18 (3.11 t·ha$^{-1}$) and Sammaz 14 (2.61 t·ha$^{-1}$) due to the variation in cultivar's maturity period and dry matter accumulation. During the testing of two maize genotypes, Cho et al. [50] found significant variations in yield i.e., 15.57% and 18.39% first and second year of experimentation, correspondingly under rainfed condition and concluded that genotype respond differentially under same condition may be due to their specific climatic requirement, and variations in growth and yield contributing characters.

Further, result (Table 5) indicated that the maximum grain yield (6197 kg·ha$^{-1}$) was recorded in N$_5$ (40 kg N as basal *fb* 2 splits of 40 kg N and 38.8 kg N at 30 and 52 DAS, respectively and 1% N foliar spray at 40 DAS) which was significantly higher than N$_3$. The data on stover yield also showed that the higher stover yield was recorded in treatment N$_5$ (11,207 kg·ha$^{-1}$) which was statistically equal to N$_1$, N$_4$ and N$_6$ but exerted significant effect over remaining treatments. The maximum grain and stover yields under treatment N$_5$ might be due to the N splitting in conjunction with foliar spray up to tasseling stage which increases leaf area, leaf area index, dry matter and yield attributes which is finally reflected with additional yield advantage. By comparing treatment N$_5$ and N$_6$ it may be concluded that Ns30RF treatments gave numerically better results till taselling stage rather than silking stage. Binder et al. [51] found that maize yields showed no positive effect of delayed nitrogen application. These results are in agreement with Neupane et al. [52] and Afifi et al. [18] who concluded that the combined approach of N management i.e., soil and foliar spray increase the yield of corn.

### 3.5. Quality Parameters

#### 3.5.1. Protein Content (%)

The data on protein content of maize grain are given in Table 6 which revealed that it was not significantly influenced by variety, although numerical more values of protein content was recorded by JM 218 (9.14%). Amongst the Ns30RF treatments, numerically higher values of protein content (Figure 8) was accumulated by N$_4$ (9.29%) followed by N$_6$ (9.28%) and N$_5$ (9.18%). It was observed that the N did not influence the protein content in tested cultivars; the results obtained might be due to ability to accumulate proteins in response to N fertilizer which depends on the cultivar [53]. Paymann [54] reported that protein content did not significantly differed due to split schedules of nitrogen application. Silva et al. [55] also reported that wheat cultivars and nitrogen scheduling did not impose its significant effect on protein content.

**Table 6.** Effect of variety and nitrogen scheduling based on 30-years average rainfall data (Ns30RF) on quality parameters.

| Treatment | Protein Content (%) | Total Minerals (%) | Dickson Quality Index |
|---|---|---|---|
| **Variety (V)** | | | |
| V$_1$- JM 216 | 9.08 [a] | 1.60 [a] | 20.60 [b] |
| V$_2$- JM 218 | 9.14 [a] | 1.63 [a] | 23.09 [a] |
| *HSD* ($p \leq 0.05$) | ns | ns | 1.85 |
| **Source of Variation** | | | |
| V | ns | ns | * |
| Ns30RF | ns | ns | ** |
| V × Ns30RF | ns | ns | ns |

Different letters, i.e., [a,b] within a column indicate significant differences according to HSD test ($\alpha$ = 0.05). Significance levels: * $p \leq 0.05$; ** $p \leq 0.01$; ns, not significant ($p > 0.05$).

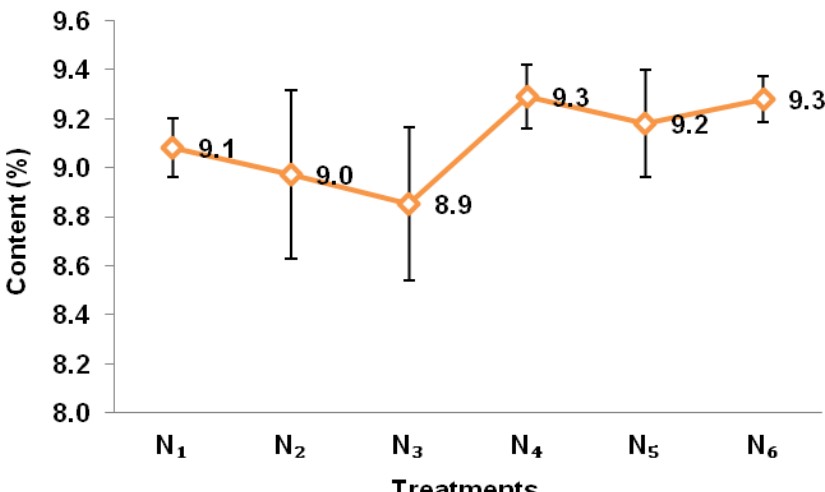

**Figure 8.** Effect of nitrogen scheduling based on 30-years average rainfall data on protein content.

3.5.2. Mineral Content (%)

The content of total minerals in maize grain has been presented in Table 6 and Figure 9. The effect of maize variety levels on total mineral content in maize grain did not bring significant changes ($p \leq 0.05$). However, the maximum total mineral content (1.63%) was reported by JM 218 variety of maize, which was 1.87% more as compared to maize variety JM 216. The levels of Ns30RF also did not show any significant ($p \leq 0.05$) role on total mineral content of maize grain. Nonetheless, the greatest amount of total mineral contents (1.63%) was accounted by $N_5$ (40 kg N as basal *fb* 2 splits of 40 kg N and 38.8 kg N at 30 and 52 DAS, respectively and 1% N foliar spray at 40 DAS), which was 2.46%, 4.40%, 5.73%, 1.84% and 1.84% higher as compared to $N_1$, $N_2$, $N_3$, $N_5$ and $N_6$, respectively. Vaswani et al. [56] concluded that different genotypes of maize differ substantially in their chemical and mineral compositions whereas some variety did not have variations in chemical and mineral compositions. Codling et al. [57] also support findings that nutrient concentrations in corn grains were not affected by the application of nutrient.

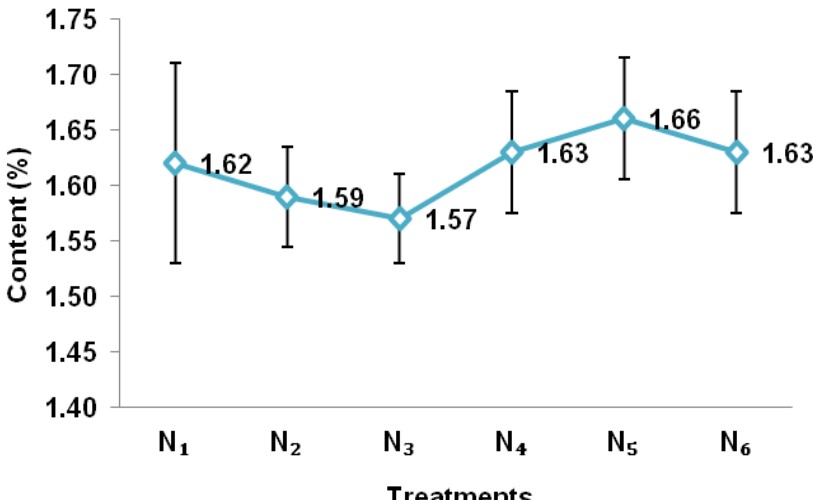

**Figure 9.** Effect of nitrogen scheduling based on 30-years average rainfall data on total mineral content.

3.5.3. Dickson Quality Index (DQI)

Maize variety JM 218 had more value of DQI (23.09) in comparison to JM 216 (Table 6). In case of various Ns30RF levels, $N_5$ (40 kg N as basal *fb* 2 splits of 40 kg N and 38.8 kg N at 30 and 52 DAS, respectively and 1% N foliar spray at 40 DAS) had significantly superior

DQI value amongst all the Ns30RF treatments (Figure 10) and exhibited its maximum effect by achieving an index value of 27.71. JM 218 and $N_5$ Ns30RF that produced more values of growth parameter and showed superiority over other treatments, which might have brought significant differences under these (JM 218 and $N_5$) treatments. Appropriate N management caused significant effect on plant height, root length, stem diameter, shoot dry matter and total dry matter which resulted in higher DQI value [58,59]. According to Dias et al. [60] suitable cultivar selection, better management of nitrogen, and increasing nitrogen doses have produced higher values of DQI.

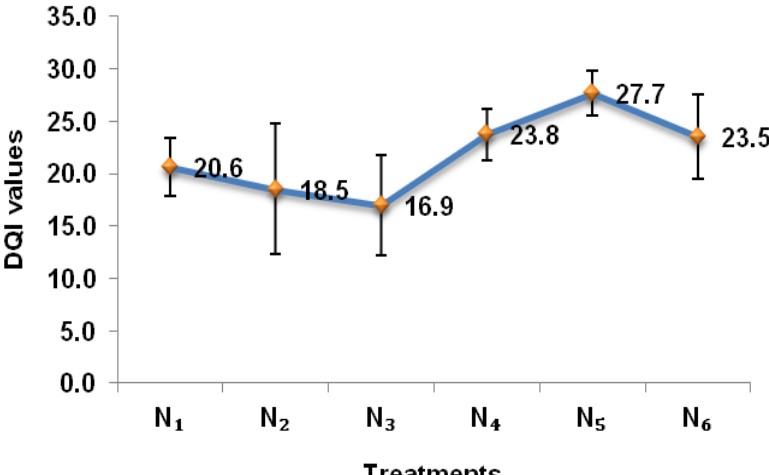

**Figure 10.** Effect of nitrogen scheduling based on 30-years average rainfall data on dickson quality index.

## 4. Conclusions

The choice of location specific variety has advantages our other varieties since it well adopted and responded better to local environment/growing conditions and applied inputs leading to significant higher production. As nitrogen is very important and primary nutrient, and its optimal availability due to proper soil profile moisture significantly enhanced growth parameters viz., plant height, dry matter, LAI and CGR which in turn improve yield and yield characteristics. The maize variety JM 218 and $N_5$ nitrogen scheduling based on 30-years average rainfall data were found promising since it recorded higher growth parameters (plant height, dry matter, LAI and CGR), yield attributes (cob length, number of grain rows·$cob^{-1}$ and test weight), grain yield (6139 kg·$ha^{-1}$ and 6197 kg·$ha^{-1}$), stover yield (11,107 kg·$ha^{-1}$ and 11,207 kg·$ha^{-1}$) and dickson quality index. Thus, based on the experimental findings it is concluded that for obtaining maximum benefits from maize cultivation under rainfed condition grower should choose JM 218 variety and apply 40 kg N as basal *fb* 2 splits of 40 kg N and 38.8 kg N at 30 and 52 DAS, in combination with 1% N foliar spray at 40 DAS based on rainfall occurrence probability because it provides better light interception by quick canopy cover, deep root system, large cob size and higher dry matter accumulation as well as higher yield and profit.

**Author Contributions:** Conceptualization and designed the experiment, A., N.S.T. and N.K.; methodology, N.S.T., N.K. and S.K.C.; software used for analysis, L.B., A.K., A.K.S. and P.S.; validation and data curation, N.S.T. and M.K.S. and R.K.S.; resources, S.K.C.; writing—original draft preparation, A., K.M., N.S.T., A.K., P.S., A.K.S. and L.B.; writing—review and editing, A., N.S.T., M.K.S., R.K.S., U.P.S., S.K.S. and N.K.; supervision, N.S.T., S.K.C., M.K.S., U.P.S. and R.K.S. All authors have read and agreed to the published version of the manuscript.

**Funding:** This research received no external funding.

**Institutional Review Board Statement:** Not applicable.

**Informed Consent Statement:** Not applicable.

**Data Availability Statement:** The data sets supporting the conclusions made are included in this article.

**Acknowledgments:** We express our sincere gratitude to Deepanita Gargav, Surbhi Sahu and Mani Yadav for the improvement of language of manuscript.

**Conflicts of Interest:** The authors declare no conflict of interest.

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
