# Peer review of "Sustainability in Rainfed Maize (Zea mays L.) Production Using Choice of Corn Variety and Nitrogen Scheduling"

_sustainability, doi:10.3390/su14053116_

Round 1

Reviewer 1 Report

Manuscript

Title: „Are 30-years long-term rainfall data based nitrogen scheduling and choice of variety brings sustainability in the production of rainfed maize (Zea mays L.)”

Authors: Aakash, Narendra Singh Thakur, Manoj Kumar Singh, Lalita Bhayal, Kamlesh Meena, Sharad Kumar Choudhary, Narendra Kumawat, Ram Kumar Singh, Udai Pratap Singh, Shrish Kumar Singh, Pratik Sanodiya, Ajay Kumar, and Anurag Kumar Singh.

Dear Authors

I revised the manuscript: " Are 30-years long-term rainfall data based nitrogen scheduling and choice of variety brings sustainability in the production of  rainfed maize (Zea mays L.)" submitted to the “Sustainability” Journal. The paper is very interesting. However, I have some concerns, which need to be addressed.

Line 5-7. Line 5-7 The title of the article generates a hypothesis. Please consider placing the hypothesis in the space of the goal and scope of the research. The title has the character of a question which conflicts with the used punctuation. The title of the article is logical but not communicative. Please pay your attention to the clarity of the statement. I suggest in this area to simplify the topic of the article to the form of a null hypothesis or to leave out the form of a question.

Abstract

Line 22 – 33. The content of the abstract is logical and understandable, the content includes basic information about the research results and conclusions.

Line 30. The abbreviation "DAS" (days after sowing) is without explanation in the summary. The explanation of the abbreviation appears first on line 142. The rather late introduction of the abbreviation explanation needs to be improved. Please include information to explain the abbreviation in the summary as well.

Line 29-30. „….40 kg N as basal fb 2 splits of 40 and 38.8kg N at 30 and 52 DAS and 1 % N foliar spray at 40 DAS…” The lack of explanation of "fb= followed" in the text and the lack of repetition of the unit of measurement for the range of values "of 40? And 38.8 kg N", creates too much confusion in the statement. Please shape the statement more clearly and understandably for the reader.

Keywords

Line 34: In my opinion, the sequence of keywords is wrong. Such words as: "corn variety", "rainfed" should be in front of other keywords. Please correct this.

Introduction

Line 35 – 105. Method and order of numbering of literature sources in the chapter correct.

The content of the chapter is logical. Informations introduces the reader well to the problems of fertilization and the influence of rain on yield dynamics.

Line 46 "CGIAR" please explain the abbreviated notation.

Line 47. please explain the abbreviation faster than in lines 477-478

Materials and Methods

2.1 Experimental site, climatic condition and soil property

Line 107 - 126 The notation of units of measurement and geographical indications is correct. The content of the chapter is logical, the informations are clearly presented without unnecessary side topics.

Line 121 - 122 Figure 1: Graphical information clearly supports the content of the subchapter.

Line 122 - 124. Figure 2. The figure supports the textual information in a highly communicative way.

Line 125 - 126. Table 1. The table correctly presents the initial information for the analyses.

2.2 Experimental set up and treatment details

Line 130 - 144 No comments to the informative value of the content and to the need to present the information in the subchapter.

Line 140 – 141. „…The hypothesis of experiment is presented in Figure A1. The image of the experiment is shown in Figure 3….” The definition of a hypothesis indicates that it is a formalised sentence. A figure cannot exist without an anticipatory sentence. A figure, in my opinion, cannot clarify a research hypothesis. The " image " of the research is the scope of the research. In my opinion, there is an overinterpretation of terms. Please take this into account.

Line 137. „….of 14-15 mm….” In my opinion the unit of measurement should be repeated for both range values. Please take this into account.

Line 143 Table 2 The scope of the study is presented in an appropriate form and is clear and understandable to the reader.

Line 144 Figure 3. The figure is readable and correctly complements the content of the subchapter.

2.3 Crop management

Line 145 - 166 The content of the chapter is understandable to the reader.

Line 149, 150, 152, 157 and similar"....@3 g...." Incomprehensible symbolism of markings, please explain this more clearly for the reader.

Line 150,151, 153, 166 and similar ".....kg ha-1....." The multiplication sign is not available - dot. Please complete the notation of the unit of measure by adding the multiplication sign.

Line 152. „….120-50-30 kg N-P2O5-K2O ha-1…..” The notation of the unit of measurement is artificially divided by the clustering of the dose terms. We do not use non-normative notations of units of measurement. This misleads the reader. Please convert the notation to the correct one.

Line 153, 154."....DAP and MOP..." please decode the abbreviated notation for better understanding of the content by readers.

2.4 Sampling and observations

2.5 Statistical Analysis  

Line 167 – 241. The description of the research methodology indicates no objections.

Line 173, 177, 191, 220, 215 and similar  „…g plant-1 …..” The multiplication sign is not available - dot. Please complete the notation of the unit of measure by adding the multiplication sign.

Line 204. „….20-30%....” In my opinion the unit of measurement should be repeated for both range values. Please take this into account.

Line 175, 232 and similar. "...68°C ± 2°C...." The unit of measure and value should be separated by a space.

Line 241."....est at 5% and 1% levels....." The notation of the percentage value with a space is incorrect. Please correct this.

Line 183, 189, 195, 210, 228, 235 The notation of mathematical formulas contains errors, incorrect notation of mathematical operation signs (colloquialisms: "x" and "÷"). Mathematical formulas do not have the necessary numbering. Please correct this.

  1. Results and discussion

Line 266, 267, 286 (Table 4), 328, 330, 333, 334, 341, 347, 350, 359, 360, 366, 370, 380 (Table 4) and similar . „….340.3 g plant-1…..” The multiplication sign is not available - dot. Please complete the notation of the unit of measure by adding the multiplication sign.

Line 270, 272. „….data 14-15 mm rainfall….” In my opinion, the unit of measurement should be repeated for all range values. Please take this into account.

Line 299, 403, 415 and similar. „….40 and 38.8 kg N…..” In my opinion, the unit of measurement should be repeated for all range values. Please take this into account.

Line 349-350 . Figure 6. The description of Figure 6 is in the wrong place. Please correct this.

Line 380-381. Table 4. I have found a table numbering error. Please correct this.

Line 404. „….2.46, 4.40, 5.73, 1.84, 1.84%.....” In my opinion, the unit of measurement should be repeated for all range values. Please take this into account.

Discussion of the results realized correctly. Authors logically justify the obtained research results.

  1. Conclusions

Line 432-442.The content of the chapter is abbreviated and narrowed only to repeat known elements from the discussion of the results. No indication of variables that might influence a different assessment of the results.

The numbering of the main chapters of the work is distorted. Please correct this.

Line 435. „….rows cob-1…..”  The multiplication sign is not available - dot. Please complete the notation of the unit of measure by adding the multiplication sign.

Line 436. „…..6139 and 6197 kg ha-1……” In my opinion the unit of measurement should be repeated for both range values. Please take this into account. The multiplication sign is not available - dot. Please complete the notation of the unit of measure by adding the multiplication sign.

Line 439."......40 and 38.8 kg N......" In my opinion the unit of measurement should be repeated for both range values. Please take this into account.

Line 459-471 Figure A1. The location of the figure is enigmatic. Please consider a different location for the graphic information.

References

Line 472-596.The selection of literature sources and the citation method are correct.

Author Response

Dear Sir/Madam,

Thank you for your time.

With best regards

Reviewer 2 Report

The paper is very clearly written and can be easily understood; still it seems to me that few and minor language and typing corrections are needed, first of all in the title. Consider asking a native English speaker to read the paper to help you improve the language.

I think DAS (days after sowing I assume) should be explained when first mentioned in the paper, for readers that are not very familiar with the subject or for ease of understanding.

When "available nitrogen" is mentioned, a short explanation is needed: I assume it is mineral nitrogen; as nitrates? as ammonium?

As you pertinently stated use of excessive nitrogen quantities for fertilization endanger environment quality. So, some research regarding environment impact of nitrogen fertilizers use in this experiment would be interesting, either completing this research or as a new one; to this end nitrates could be determined in soil at the end of the experiment, in the edible parts of the crop (as nitrates are known to negatively affect the consumers’ health), and in the groundwater which mainly contributes to nitrates spreading in the environment.

Author Response

(The authors gave the same response as above.)

Reviewer 3 Report

This manuscript presents an interesting and beneficial case study. Additionally, the theme of the paper is within the scope of the journal. The paper is technically correct, well organized. The illustrations are generally adequate and the results look secure.

Author Response

Dear Reviewer,

Thank you for your time.

With best regards
